# In Silico Screening and Optimization of Cell-Penetrating Peptides Using Deep Learning Methods

**DOI:** 10.3390/biom13030522

**Published:** 2023-03-13

**Authors:** Hyejin Park, Jung-Hyun Park, Min Seok Kim, Kwangmin Cho, Jae-Min Shin

**Affiliations:** 1RM 101-1702 ADLi Institute, AZothBio. Inc., 109 Mapo-daero, Mapo-gu, Seoul 04146, Republic of Korea; 2RM D-724 Hyundai Knowledge Industry Center, AZothBio. Inc., 520 Misa-daero, Hanam-si 12927, Republic of Koreakmcho@azothbio.com (K.C.)

**Keywords:** cell-penetrating peptides, CPP, deep learning, drug delivery system

## Abstract

Cell-penetrating peptides (CPPs) have great potential to deliver bioactive agents into cells. Although there have been many recent advances in CPP-related research, it is still important to develop more efficient CPPs. The development of CPPs by in silico methods is a very useful addition to experimental methods, but in many cases it can lead to a large number of false-positive results. In this study, we developed a deep-learning-based CPP prediction method, AiCPP, to develop novel CPPs. AiCPP uses a large number of peptide sequences derived from human-reference proteins as a negative set to reduce false-positive predictions and adopts a method to learn small-length peptide sequence motifs that may have CPP tendencies. Using AiCPP, we found that short peptide sequences derived from amyloid precursor proteins are efficient new CPPs, and experimentally confirmed that these CPP sequences can be further optimized.

## 1. Introduction

In drug delivery, intracellular delivery of therapeutic drug molecules is essential. For drugs to exert their therapeutic effect in intracellular organelles such as the cytoplasm, nucleus, lysosomes, and mitochondria, they must cross the cell membrane. However, hydrophilic therapeutics or large biomolecules, such as proteins or nucleic acids, cannot typically penetrate cells. Cell-penetrating peptides (CPPs) are typically 4–40 amino acids in length, and they can penetrate cells to enhance the cellular uptake of various molecular cargoes [1,2,3,4,5,6,7]. CPPs have also been the focus of increased research interest because they are non-toxic and do not permanently damage cell membranes upon entry [7,8].

Remarkable progress has been made in the development of CPP-based drug delivery systems in various preclinical studies over the past decade [9,10,11,12,13]. Hundreds of novel CPPs and chemically modified analogs have been discovered, and several CPP-based formulations are currently being evaluated for toxicity and safety in various stages of clinical trials [7,8]. Although several efficient CPPs have been identified, they do not have a wide range of possible applications. Consequently, their improvement and the discovery of new and more efficient CPPs with increased bioavailability and reduced side effects for safe drug delivery and therapeutic applications remains a challenge [14,15,16].

Experimental verification and optimization of CPPs is time consuming and labor intensive, therefore, various in silico methods for CPP prediction have been devised as alternative approaches [17,18,19,20,21,22,23,24,25,26,27,28]. Mining properties of peptides such as amino acid composition, biochemical properties, and methods of expression of numerous new functions have been used in several in silico CPP prediction methods to achieve more than 80% accuracy. Researchers have made significant progress in developing new prediction algorithms by utilizing the physicochemical properties or distance information of a peptide sequence. For instance, CellPPD [19] and machine learning-based prediction of CPP (MLCPP) [20] use various features such as amino acid composition, dipeptide composition, a binary profile of patterns, and physicochemical properties as input features for CPP prediction. Recently, CPPpred [21] utilizes an N-to-1 neural network to express the peptide sequence as a feature of a fixed length.

However, current methods, including these studies, still have several problems that could be improved to make them more useful for CPP research. For example, they are based on relatively short CPP peptide sequences, which can make it difficult to pinpoint which parts of a protein or peptide sequence have CPP properties, and they are trained on a limited number of CPP data, which can lead to a large number of false-positive CPP predictions.

In this study, we propose AiCPP, a deep learning-based model for predicting cell permeation propensities of peptide sequences. In the development of AiCPP, we used a large number of negative CPP dataset generated from a human reference protein sequence database to reduce false positives and 9-mer peptide sequences as input to learn local-CPP sequence patterns more efficiently. The AiCPP was able to significantly reduce false-positive predictions and show more robust CPP sequence learning and prediction results.

As a model case for experimental performance verification of AiCPP, we predicted and selected a novel CPP sequence from amyloid precursor protein (APP) sequence, and confirmed that it is a good CPP through a cell permeability test in MCF-7 cells. Furthermore, we found that AiCPP can be used to optimize the wild-type APP-derived CPP sequence into a CPP sequence with higher CPP efficiency.

Figure 1 shows the schematic diagram describing the processes used to build the AiCPP model.

## 2. Materials and Methods

### 2.1. CPP Dataset Preparation

The data generation process is shown in Figure 2. We collected data for known cell-penetrating peptides (CPPs) from several sources, including CellPPD [19], MLCPP [20], CPPsite 2.0 [29], Lifetein (https://www.lifetein.com/ (accessed on 07 December 2021)), and other publications [4,17,27]. From this collection, we selected 2798 unique peptides that were between 5 and 38 amino acids long, after removing redundant peptides and those with non-amino acid sequences. To test the performance of our model, we created a separate set of 150 CPP peptides and 150 non-CPP peptides that were not used in the other three models (MLCPP, CellPPD, CPPred). In total, we used 2346 peptides for the train set, including 1249 CPPs and 1097 non-CPPs.

To prepare the input data for our deep learning model, we used the sliding window method to slice the peptide sequences into overlapping 9-amino acid segments, as shown in Figure 3. Using the sliding window method to slice the curated peptides into overlapping 9-amino acid segments allows us to use more training data and capture local sequence patterns or meaningful sequence context features. The sliding window approach is commonly used in molecular sequence analysis to study the properties of individual residues.

To ensure that all peptides were of uniform length, we padded shorter sequences with ‘-’ characters to create 9-mer peptides. This step was necessary to maintain consistency across the dataset for deep learning. We removed all duplicate 9-mer peptide sequences from the dataset.

We generated 11,046,343 9-mer peptide fragments from 113,620 human reference proteins to be used as the negative set in the training process. By including a large number of negative datasets, we hoped to improve the model’s specificity, or its ability to correctly identify non-CPPs, by reducing the bias toward predicting false positives.

Finally, after removing duplicates in 9-mers, the AiCPP model was trained on 21,573 peptide fragments, including 7165 positive (CPP) 9-mer peptides, 14,408 negative (non-CPP) 9-mer peptides, and 11,046,343 negative 9-mer peptides derived from human reference proteins (Figure 2).

### 2.2. Model Algorithms and Assessment

The AiCPP uses ensemble learning, which involves training multiple models and combining their predictions to obtain a more accurate and robust result. Specifically, we built five different deep-learning architectures, each including an embedding layer, a long short-term memory (LSTM) layer, and attention layers. Appendix A and Table 1 show more detailed information about these architectures.

To generate the input for the model, we convert the peptide sequence into a dense vector using an embedding layer. The resulting vector is used as the input for each of the five models, which use the binary cross entropy loss function and are trained for 1000 epochs using the Adam optimizer.

To obtain a final prediction value for a given peptide sequence, we take the average of the prediction values of each 9-mer obtained using a sliding window approach. Our model was implemented using Python 3.8 and TensorFlow 2.4.0.

Several metrics were used to evaluate the performance of the AiCPP model, including area under the curve (AUC), accuracy (ACC), sensitivity (SEN), specificity (SPE), precision (PRE), and Matthew’s correlation coefficient (MCC).

The area under the curve (AUC) measures the ability of the model to discriminate between positive (CPP) and negative (non-CPP) samples. We calculated it by plotting the true positive rate (sensitivity) against the false positive rate (specificity) for a range of classification thresholds and then calculating the area under the resulting curve. A higher AUC value indicates a better performing model.

Accuracy (ACC) measures the proportion of correct predictions made by the model. We calculated it as the number of correct predictions divided by the total number of predictions.

Sensitivity (SEN) measures the proportion of true positive predictions made by the model. We calculated it as the number of true positive predictions divided by the total number of actual positive samples.

Specificity (SPE) measures the proportion of true negative predictions made by the model. We calculated it as the number of true negative predictions divided by the total number of actual negative samples.

Precision (PRE) measures the proportion of correct positive predictions made by the model. We calculated it as the number of true positive predictions divided by the total number of predicted positive samples.

Matthew’s correlation coefficient (MCC) measures the ability of the model to correctly predict positive and negative samples. We calculated it as the product of sensitivity and specificity divided by the square root of the product of the true positive rate, the true negative rate, the false positive rate, and the false negative rate. A higher MCC value indicates a better performing model [30].

### 2.3. Calculation of Cell-Penetrating Propensity for Each Residue in the Peptide Sequence

To calculate the cell-penetrating propensity of each amino acid in the sequence, we took the average score of overlapping 9-amino acid segments that contain the specific amino acid. Figure 4 shows an example of this calculation, where the cell-penetrating propensity of the Asp (D) residue in the red box is represented by the average of the predicted values of nine peptide fragments containing Asp.

### 2.4. Discovery and Optimization of CPP

We validated the AiCPP model using a 770-amino acid long amyloid precursor protein (APP) as a test case. Using five different models (Model 1 to Model 5), we predicted the cell-penetrating propensity of the APP protein using the AiCPP model. After analyzing the prediction results, we identified the signal peptide, a 17-amino acid region of the APP protein, as the wild-type (WT) CPP.

To find more optimized CPPs within the WT CPP, we generated a sequence space of approximately 4.71 million peptides by substituting up to three amino acids in the WT CPP. We then calculated the CPP prediction scores for each peptide using the AiCPP model. Our goal was to identify the most efficient CPPs, so we selected nine optimized peptides with a higher cell-penetrating propensity than the WT CPP based on the predicted values.

### 2.5. Peptide Synthesis

Ten peptide sequences explained in Section 2.4 were synthesized by Fmoc-based solid phase peptide synthesis using an automated ASP48S peptide synthesizer (Peptron, Daejeon, Republic of Korea) and are listed in Table 2.

All peptides are modified with N-terminal fluorescein isothiocyanate (FITC) and C-terminal amidation. The synthesized peptides are purified by reversed-phase high-performance liquid chromatography using a Vydac C18 column and gradient elution with a mixture of water/acetonitrile containing 0.1% trifluoroacetic acid. The molecular weight of each peptide is estimated and confirmed using mass spectrometry (MS) analysis using a liquid chromatography-MS (LCMS)-2020 system (Shimadzu, Kyoto, Japan) (Table 2).

### 2.6. Cell Culture and Peptide Treatment

MCF-7 cells are maintained in Roswell Park Memorial Institute 1640 medium supplemented with 10% fetal bovine serum and 1% antibiotics at 37 °C in an atmosphere of 5% CO_2_ in the air. Cells are cultured in a 35-mm plate at a density of 5 × 10^5^ cells in growth medium, incubated overnight, and then treated with 1 µM FITC-conjugated peptide for 1 h at 37 °C in a CO_2_ incubator.

### 2.7. Cell Permeability Assay

The cultured cells are harvested, washed twice with phosphate-buffered saline (PBS), treated with trypsin for 5 min, washed with PBS again, and then resuspended in 1 mL of 2% paraformaldehyde in PBS. Cells were then transferred to a fluorescence-activated cell sorting tube and analyzed by flow cytometry using a Novocyte flow cytometer (Agilent, Santa Clara, CA, USA).

## 3. Results

### 3.1. Performance of AiCPP in CPP Prediction

The AiCPP model is an ensemble model that combines the predictions of five different classification models, labeled Model 1 through Model 5. These models have accuracies ranging from 0.823 to 0.860, which are not significantly different from each other (Table 3).

However, the predictions of each model for the same peptide can vary. A pair plot (Figure 5) illustrates the differences in prediction values between the models for the same peptide. In this figure, the dots that are far from the x = y region represent the different predictions made by the models. This indicates that although some peptides are well predicted by each model, the predictions between them may differ slightly.

To compensate for the individual weaknesses of the models and improve the overall performance of the AiCPP model, the ensemble averaging method is used. This combines the predictions of the five models to produce a final prediction score. Ensemble averaging takes into account the differences in the predictions made by each model and improves the accuracy of the final prediction. This is reflected in the significantly increased AUC and MCC values for the AiCPP model, indicating a better performance than any individual model alone.

Table 3 shows that the AiCPP model outperforms the other models in terms of AUC, MCC, and ACC. When evaluated using a test set, the AiCPP model outperforms MLCPP, CellPPD, and CPPred, with values of 0.927, 0.722, and 0.860 for AUC, MCC, and ACC, respectively. In addition, the AiCPP model has a higher specificity for non-CPPs (0.893) compared to the three external models listed in Table 3. Overall, these results demonstrate that the AiCPP model is a highly accurate and reliable method for predicting CPPs.

The improved specificity of AiCPP can be attributed to the use of a large number of negative datasets derived from human reference proteins, which reduces the number of false positives compared to other models. This feature makes AiCPP a valuable tool for large-scale CPP screening, as it can significantly reduce the potential number of CPPs for experimental validation.

### 3.2. CPP Screening and Optimization Using AiCPP for Human APP Protein

The AiCPP model was utilized to identify potential cell-penetrating peptides (CPPs) in the human amyloid precursor protein (APP), a 770-amino acid protein that is sometimes mutated, leading to the aggregation of amyloid beta and the development of Alzheimer’s disease [31]. Using the sliding window method, the AiCPP model calculates the cell penetration propensity of each sequence in APP. The results show that sequences 1–17 (MLPGLALLLLAAWTARA) and 93–108 (TIQNWCKRGRKQCKTH) have a high probability of being CPPs (Figure 6). Of particular interest is sequence 1–17 (WT CPP), which functions as a signal sequence to transport APP to the endoplasmic reticulum and was selected for further experimental validation of the AiCPP model.

To identify a more optimized CPP with improved cell permeability based on the WT CPP, a sequence space consisting of 4.71 million sequences, including up to three mutations in the WT CPP sequence, was searched. Figure 7 shows the histogram of the AiCPP scores. Of the 4.71 million sequences generated, the AiCPP model predicted 5948 sequences with a score ≥ 0.8.

Figure 8 shows the position-specific amino acid propensity scores calculated by AiCPP for the CPP. The logo plot shows the amino acid occurrence rate at each position along the *X*-axis, while the *Y*-axis indicates the frequency of amino acid occurrence. Sequences with a predicted AiCPP value of 0.8 or greater exhibit a high mutation frequency at positions 3, 4, and 14, corresponding to P, G, and T, respectively.

We selected 10 CPP candidates, listed in Table 2, based on the sequence logo plot (Figure 8) and AiCPP scores. Peptide 1 is the WT CPP sequence from the amyloid precursor protein, whereas peptides 2–10 are peptides optimized by AiCPP. These peptides were generated by modifying the 3rd, 4th, or 14th position of the WT peptide and replacing the amino acids with K, R, or H, as indicated by the sequence logo plot. The optimized peptides show significantly increased AiCPP scores compared to the WT peptide, with the score of peptide 9 reaching 0.886 (Figure 9). The AiCPP score of the WT peptide was 0.542.

### 3.3. Enhanced Cell-Permeability of Modifided Peptides in MCF-7 Cells by AiCPP Optimization

In our experiments, we tested the cell permeability of the wild-type peptide and of nine optimized peptides in MCF-7 cells. As shown in Figure 10, peptide 3–9 exhibited significantly improved permeability compared to the WT peptide (Peptide 1), which had modifications at the 3rd, 4th, and 14th positions, as identified by the sequence logo plot (Figure 8). These modifications involved the substitution of certain amino acids with K, R, or H, which increased the cell-penetrating peptide (CPP) ability of the peptides.

However, peptide 2 and peptide 10, which were modified at the 2nd or 13th position, exhibited lower cell permeability than the WT CPP (Figure 10). The results suggest that simply replacing any position with K, R, or H does not necessarily increase the cell permeability, but the sequence context is more important.

## 4. Discussion

Machine learning is increasingly used to discover more efficient CPPs, but current techniques face limitations due to insufficient and diverse data. Most existing models [19,20,21] are trained on fewer than 1000 CPPs, which limits extrapolation of all possible sequences of similar length. To compensate for the lack of experimentally validated non-CPP data, many models use randomly generated sequences as negatives [19,20].

In this study, we adopted a sliding window approach that divided the collected CPP data into nine amino acid segments, effectively increasing the amount of data available for CPP training by about 10-fold. In addition, we reduced the number of false positives in the CPP prediction model by adding sequences with low similarity to CPP sequences in the human reference protein sequence as a negative set. The effect of using a large number of negative datasets is obvious. When we trained and tested the AiCPP model without using this large number of negative sets, the metrics for the measure of false-positive predictions, such as MCC and SPE, were significantly reduced (see Appendix A). Interestingly, despite using 11,046,343 negative 9-mer data from human reference proteins, the AiCPP predicted that 770,435 9-mer peptides (about 6.97%) were predicted as positive CPP (see Appendix A).

We also utilized the amino acid sequence context information of the CPP sequence itself, which is commonly used in deep learning techniques such as LSTM and attention methods, to learn and predict CPPs. This 9-mer sequence context-based approach resulted in good CPP prediction performance compared to previous methods. With this approach, we obtained improved results and hope that it can be used to identify more efficient CPPs and further contribute to a more comprehensive understanding of CPPs based on the sequence information of CPPs and non-CPPs. As an experimental test case, we found a CPP sequence predicted by AiCPP in the APP protein sequence and confirmed that the sequence predicted by AiCPP was a novel CPP in MCF-7 cells. Furthermore, we found that AiCPP can be used to optimize CPP sequences with higher CPP efficacy.

While this study and many previous in silico CPP prediction studies [17,18,19,20,21] have shown great progress, they have not answered many important questions about the mechanisms of cellular permeation and how each CPP sequence is important for cellular permeation [32]. Furthermore, CPP permeability is known to be cell type dependent [33], so it is unreasonable to extrapolate the results of the AiCPP optimization sequence and cell permeability experiments from one cell type, such as MCF-7, used in this study. However, as more data becomes available for different cell types, in silico machine-learning-based CPP detection methods, such as AiCPP, may offer promising opportunities for future research.

It is also known that some CPPs can self-assemble into oligomers and promote cellular uptake. However, the mechanism of formation of CPP oligomers and their relationship with monomers in cellular penetration are not yet fully understood [34]. Although CPP peptides predicted by AiCPP would have a high propensity to enter cells, it is still unclear whether these CPPs, including the APP-derived CPPs we found in this study, enter cells as monomers or oligomers. We believe that further studies are needed to address this issue.

In summary, we present a novel in silico method for the discovery of cell-penetrating peptides (CPPs) that incorporates deep learning-based natural language processing techniques, such as long short-term memory (LSTM) and attention mechanisms, in a sliding window approach. The results show that this approach can effectively overcome the limitations of current machine learning methods for CPP discovery. The study also provides valuable insights into the sequence patterns required to optimize CPPs, which can inform the design of novel CPPs with enhanced cell penetration properties. By utilizing deep learning-based techniques and the sliding window approach, our study demonstrates an improved ability to predict CPP sequence patterns and identify novel CPPs with superior cell penetration properties. These results have important implications for the development of future CPP-based therapies and highlight the potential of our in silico approach to accelerate the discovery of effective CPPs.

## Figures and Tables

**Figure 1 biomolecules-13-00522-f001:**
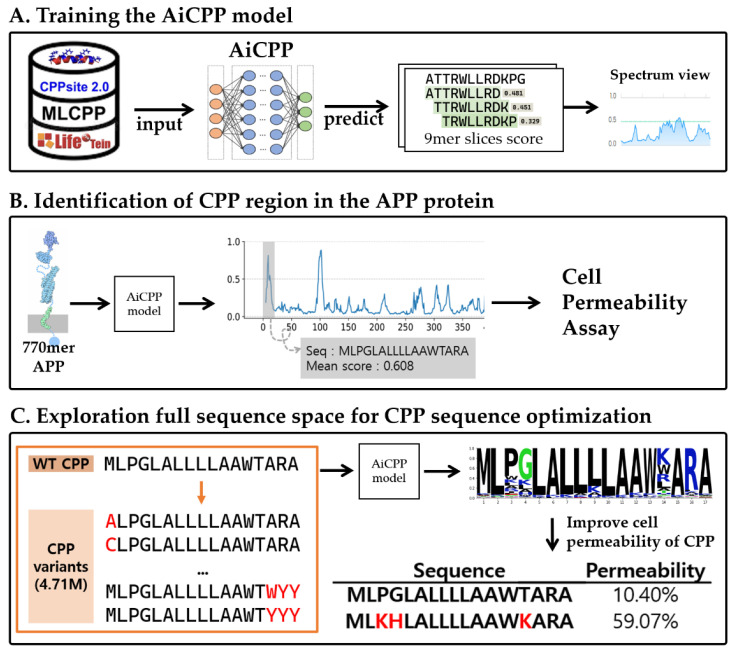
Schematic workflow of the AiCPP, showing (**A**) training the AiCPP model, (**B**) identification of the CPP region in the APP protein, and (**C**) exploration of full sequence space for CPP sequence optimization.

**Figure 2 biomolecules-13-00522-f002:**
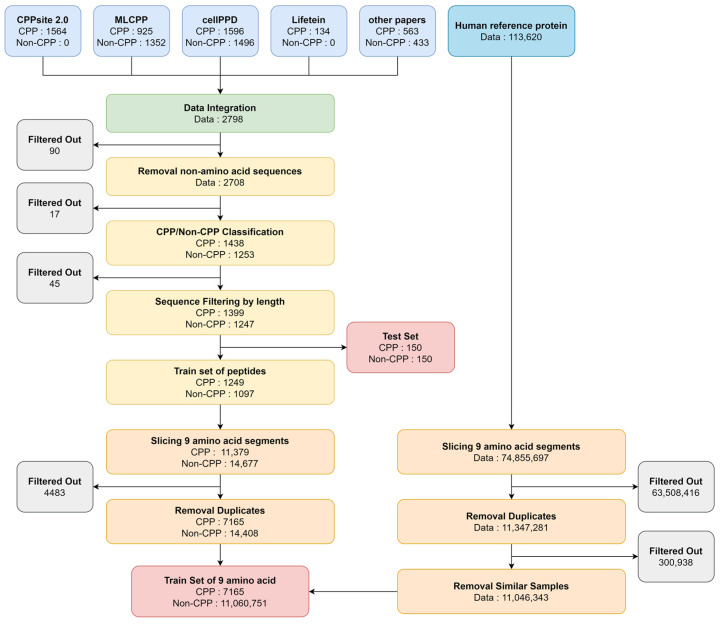
Dataset preparation. Peptide sequences derived from the human reference protein are used as the negative set.

**Figure 3 biomolecules-13-00522-f003:**
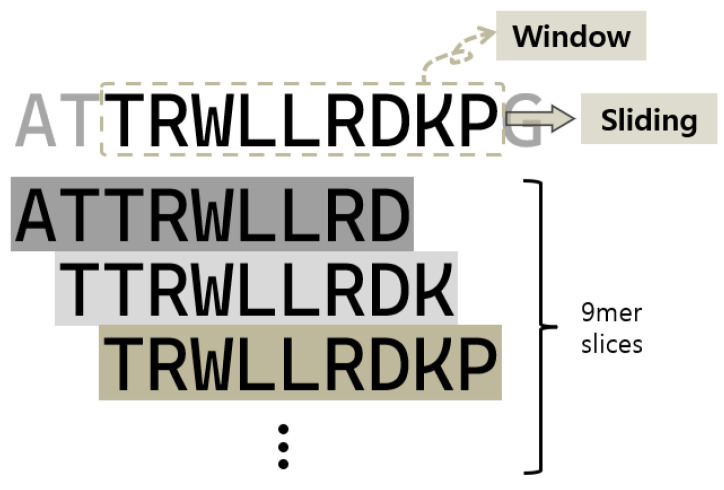
Preparation of 9-mer peptide sequences using the sliding window method for training the AiCPP model.

**Figure 4 biomolecules-13-00522-f004:**
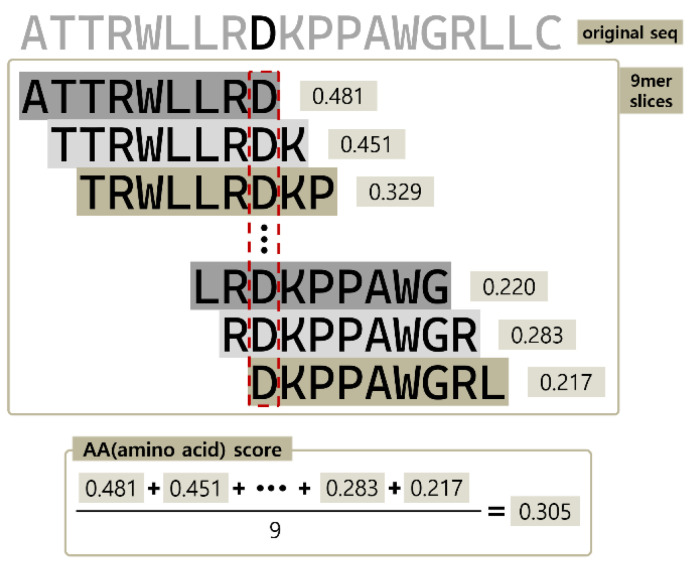
Calculation of Residual cell-penetrating propensity.

**Figure 5 biomolecules-13-00522-f005:**
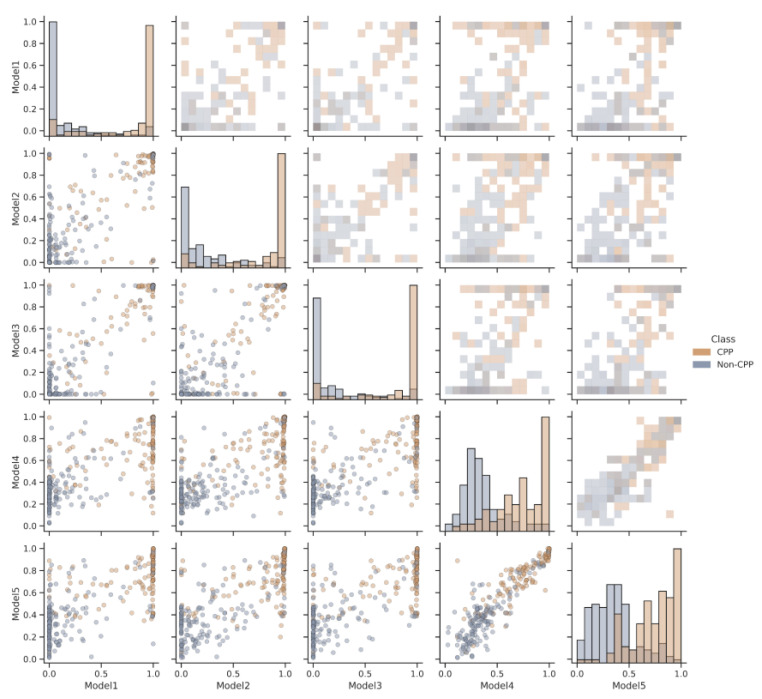
Pair plot of the AiCPP models’ correlation. Each dot in the two-dimensional scatterplot represents a peptide, either CPP (orange) or non-CPP (blue), and compares the predicted values of each model.

**Figure 6 biomolecules-13-00522-f006:**
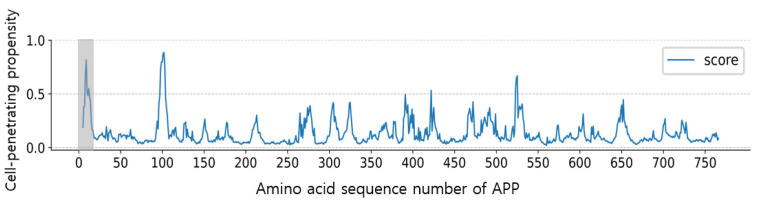
Cell-penetrating propensity of residues in the APP protein predicted by the AiCPP model, with the identified CPP peptides (1–17) highlighted in a gray box.

**Figure 7 biomolecules-13-00522-f007:**
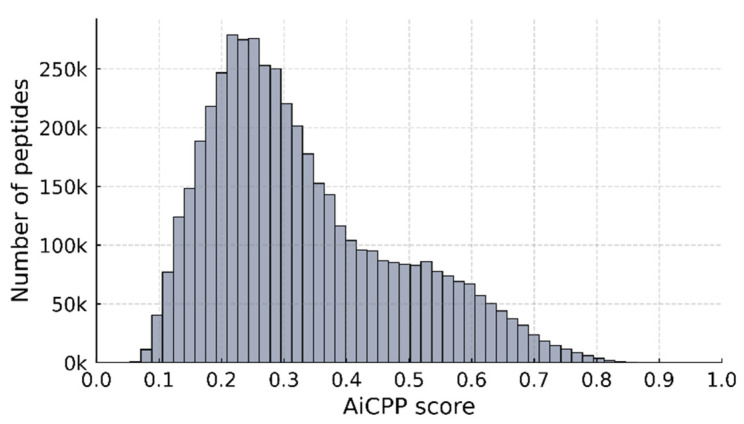
Histogram of AiCPP scores for mutant peptides predicted by the AiCPP model. The *X*-axis represents the AiCPP score, and the *Y*-axis represents the number of peptides with that AiCPP score.

**Figure 8 biomolecules-13-00522-f008:**
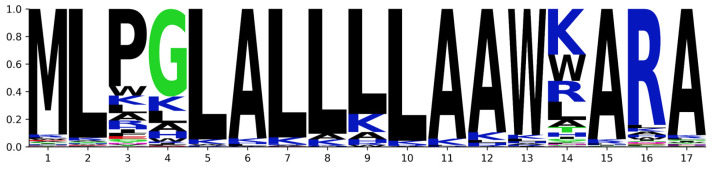
Sequence logo plot of 5948 mutant sequences with a predicted AiCPP value of 0.8 or higher. The *X*-axis denotes the position of the amino acid, and the *Y*-axis represents the frequency of the amino acid appearance rate.

**Figure 9 biomolecules-13-00522-f009:**
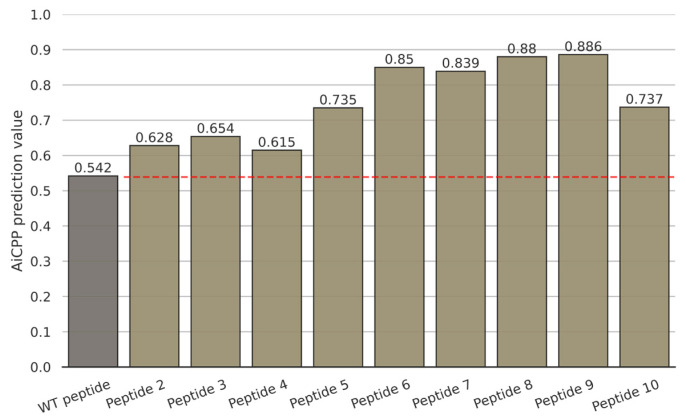
AiCPP prediction values for nine optimized peptides. The bar graph shows the AiCPP scores, with the red dashed line indicating the score of the WT peptide.

**Figure 10 biomolecules-13-00522-f010:**
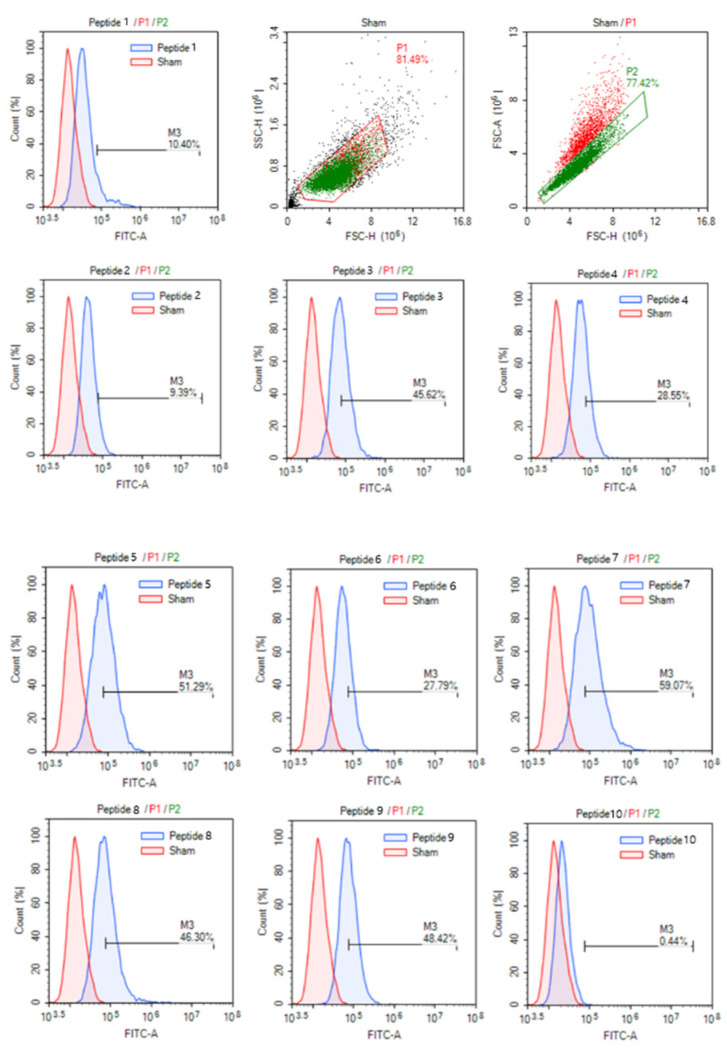
Cellular uptake of CPPs. Cells are treated without (sham) or with each FITC-conjugated peptide (1 µM) for 1 h in MCF-7 cells, and the fluorescence intensity is measured using flow cytometry as described in the Materials and Methods section.

**Table 1 biomolecules-13-00522-t001:** Five different model architectures used in AiCPP.

	Embedding Dimension	Convolution Layer	LSTM Layer	Attention Layer	Parameters
Model 1	10	15	2	3	11,386
Model 2	10	15	0	2	5218
Model 3	10	15	3	6	7405
Model 4	3	0	0	1	1275
Model 5	6	0	0	1	1054

**Table 2 biomolecules-13-00522-t002:** Ten CPP-peptide sequences synthesized. Peptide 1 is the WT peptide.

Name	Sequence	*m*/*z*	Ion	Molecular Formula
Peptide 1	N′-MLPGLALLLLAAWTARA-C′	2282.76	(M + H)^+^	C_111_H_164_N_24_O_24_S_2_
Peptide 2	N′-MLPGLALLKLAAWTARA-C′	2297.83	(M + H)^+^	C_111_H_165_N_25_O_24_S_2_
Peptide 3	N′-MLPGLALLLLAAWKARA-C′	2309.89	(M + H)^+^	C_113_H_169_N_25_O_23_S_2_
Peptide 4	N′-MLPGLALLLLAAWRARA-C′	2337.84	(M + H)^+^	C_113_H_169_N_27_O_23_S_2_
Peptide 5	N′-MLKGLALLLLAAWKARA-C′	2340.89	(M + H)^+^	C_114_H_174_N_26_O_23_S_2_
Peptide 6	N′-MLPKLALLLLAAWKARA-C′	2380.95	(M + H)^+^	C_117_H_178_N_26_O_23_S_2_
Peptide 7	N′-MLKHLALLLLAAWKARA-C′	2420.97	(M + H)^+^	C_118_H_178_N_28_O_23_S_2_
Peptide 8	N′-MLAKLALLLLAAWKARA-C′	2354.91	(M + H)^+^	C_115_H_176_N_26_O_23_S_2_
Peptide 9	N′-MLKKLALLLLAAWKARA-C′	2412.01	(M + H)^+^	C_118_H_183_N_27_O_23_S_2_
Peptide 10	N′-MKPGLALLLLAAKKARA-C′	2266.81	(M + H)^+^	C_108_H_172_N_26_O_23_S_2_

**Table 3 biomolecules-13-00522-t003:** Performance evaluation of the CPP predictors. The best performance metrics are in bold.

	AUC	MCC	ACC	SEN	SPE	PRE
Model 1	0.893	0.711	0.853	0.800	**0.907**	**0.896**
Model 2	0.878	0.647	0.823	0.827	0.820	0.821
Model 3	0.878	0.674	0.837	0.813	0.860	0.853
Model 4	0.879	0.660	0.830	0.827	0.833	0.832
Model 5	0.891	0.660	0.830	0.827	0.833	0.832
AiCPP (ensemble)	**0.927**	**0.722**	**0.860**	0.827	0.893	0.886
MLCPP	0.882	0.633	0.810	**0.914**	0.705	0.758
CellPPD	0.724	0.452	0.723	0.656	0.792	0.762
CPPred	0.845	0.564	0.780	0.722	0.839	0.820

## Data Availability

Not applicable.

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
