# Peer review of "In Silico Screening and Optimization of Cell-Penetrating Peptides Using Deep Learning Methods"

_biomolecules, 2023, doi:10.3390/biom13030522_

Round 1

Reviewer 1 Report

Please explain how you select the number of samples in each level. 
Please send raw data of models 
Please send a log book of cell culture and cell permeability. 
Please correct these words. 
Analogs (35)
Processes (66)
Contains (82)
Training of the model (98)
A long short term (110)
Are > is (138)
"," > without "," (209)
Calculates (252)
The amyloid (281)
Was predicted ( 301)

Author Response

Please see the attachement. Thank you.

Reviewer 2 Report

 Park et al. describe the development of a novel machine-learning based protocol for the prediction of cell penetrating peptides. I believe that the paper has merit and eventually deserves to be published, but there is a number of things, if addressed, will make this a much stronger paper. Below are my comments on things that can be improved:

·         Is the permeation mechanism reliant on a single peptide or does it involve peptide oligomerization?

·         How can it be argued that the 11 million peptides taken from human sequences are not going to be membrane penetrating? The 9-mer sequences may in fact be membrane permeable in the form of free peptides although they may not be able to be so in the context of a globular protein. I think one way to measure this would be running AiCPP on the human protein sequence sets to see whether there are any sequences predicted to be membrane-penetrating by the algorithm itself and comment on their statistics.

·         The authors should comment on the conditions under which these peptides penetrate the cells. The data come from multiple databases, can the cell types these experiments were done with be compared? For example, are there any cases where the data were collected from cells with negatively-charged membranes as opposed to positively charged or neutral membranes?

·         The synthesized peptides should be tested with the other methods available for CPP prediction. It is hard to comment on the success of the AiCPP method in designing peptides otherwise.

·         It would be nice to see the structural effects of the mutations. The authors should model the synthesized peptide structures to see whether the mutant effects are caused by structural changes or due to changes in the specific interactions with the cell membranes. There are multiple peptide modeling servers for this purpose, or ML tools like AlphaFold or OmegaFold (which are more accurate) can be used to run such calculations within a reasonable amount of time.

·         Once the structures are modeled it would be very impactful to see the energetics of membrane penetration for the different peptides. I would consider this totally optional, but it would be a great exercise to computationally calculate the energy of the peptides throughout an explicit or implicit membrane model and see how these individual mutations might affect penetration in reality.

·         The “average score” approach may skew the actual effect of the active regions. I feel that what determines the cell penetration of a peptide would be its most permeable region rather than its whole sequence. How is the model performance affected if training is done only with the regions predicted to have highest penetration?

Author Response

Thank you.

Reviewer 3 Report

In this work, the authors attempted to use a new machine learning approach to decode and then optimize the ability of the cell-penetrating peptides.  It is a strong paper in all aspects to this reviewer who believes that it will be publication ready after addressing the following rather minor points.

11. The authors mentioned that they used a sliding window approach to divide the collected peptides into overlapping 9-amino acid segments.  However, they also mentioned in manuscript line 85 that the collected peptides range in length from 5 to 38 amino acids.  What happened to those with less than 9 amino acids as they apparently do not fit in this method.  This should be clarified.   

22. The authors emphasized that the inclusion of a negative set improves their approach by reducing the probability of false positive.  This makes sense from a fundamental point of view and can be or may have been adopted by other training methods. Judging by its importance, it would be desirable if the authors can provide some comment plus references or some discussion.      

33. While enjoying reading this manuscript, this reviewer kept thinking a couple of questions related to the anchor/hypothesis of the authors’ approach, that is, the cell-penetrating ability of peptides can be attributed to and hence estimated based on that of the constituent amino acids (e.g., Figs. 3 and 5).  If so, wouldn’t it imply that assembling the same high scoring amino acids into a homopolypeptide can produce a highly cell-penetrating peptide?  Furthermore, exchanging amino acids among themselves in a peptide without changing the types or numbers of amino acids would have the same cell-penetrating propensity in the authors’ approach but such changes of amino acid sequences are known to cause the peptides to have different properties and thus would not be expected automatically to have the same cell-penetrating propensity.  This review tends to think that many readers may have the same questions and thus believe that these points should be addressed.            

44. This reviewer was somewhat misled by the phrases ‘“complex patterns” of short peptides’ and ‘“complex patterns” hidden in the short CPP sequence’, especially this word “complex” because the manuscript did not really hint any complexity of short peptides.  I think the authors meant a large number of possible mutations/variations.  Would it be beneficial to replace the misleading phrase of “complex patterns”?

55. Section 4 is titled “Discussion” but it reads like “Conclusions”.  Can the authors review and evaluate again to see which one is more proper?    

66. What is LSTM in Table 2? 

77. There are repetitions, even in the same paragraphs, for example, between manuscript line 74 and line 86, between line 92 and line 102, and between line 296 and line 306.  Also, the content of the paragraph between line 146 and line 155 has been detailed previously in Section 2 and Fig. 2.  In addition, line 303 reads “change to K, R, or H at any location on the peptide matched the AiCPP prediction in the cell” which is not clear what exactly the authors mean.  It would be desirable if the authors can review and perhaps rewrite these parts.      

Author Response

Thank you.

Round 2

Reviewer 2 Report

The authors have done an excellent job in responding my comments. I think the article is publishable in this form.